# Characterization and Fungicide Screening of a New Pathogen That Causes Leaf Spot on *Rehmannia glutinosa*

Tao Dou [1,†], Yubiao Cai [1,†], Xuhong Song [2], Futao Gao [1], Yajun Zhao [1], Jiafang Du [1], Fengqing Wang [1], Xuanzhen Li [1], Shiheng An [1], Xinming Yin [1], Xiangyang Liu [1,*] and Zhongyi Zhang [3]

1    Wenhua Road Campus, Henan Agricultural University, Zhengzhou 450002, China
2    Chongqing Academy of Chinese Materia Medica, Chongqing 400065, China
3    College of Agriculture, Fujian Agriculture and Forestry University, Fuzhou 350002, China
*    Correspondence: liuxiangyang@henau.edu.cn
†    These authors contributed equally to this work.

**Abstract:** Outbreaks of leaf spot disease occurred in *Rehmannia glutinosa* fields in Henan Province, China, in 2019, with the incidence ranging from 20% to 40%. *R. glutinosa* plants with diseased leaves were collected, and 25 isolates were obtained. Pathogenicity tests, morphological observations, and phylogenetic analyses were conducted to identify the pathogens, and the biological characteristics and control agents of the pathogens were studied. Five isolates of pathogenic fungi were isolated. Three isolates were identified as *Fusarium equiseti*, which is a new pathogen causing *R. glutinosa* leaf disease; the other two isolates were identified as *Fusarium acuminatum*. The mycelia of *F. equiseti* grew fastest on Czapek medium, and the optimal temperature and pH were 25 °C and 10.0, respectively. The mycelia of *F. equiseti* grew from 5 °C t o 35 °C, and the lethal temperature was 55 °C. The optimal carbon and nitrogen sources were soluble starch and peptone, respectively. Eight fungicides had inhibitory effects on the mycelial growth of *F. equiseti* and *F. acuminatum*. Prochloraz had higher activities against *F. equiseti* and *F. acuminatum*, with $EC_{50}$ values of 0.139 mg·L$^{-1}$ and 0.123 mg·L$^{-1}$, respectively. These results provide useful information that will aid the development of management strategies to control leaf diseases of *R. glutinosa* caused by *F. equiseti* and *F. acuminatum*.

**Keywords:** *Rehmannia glutinosa*; disease; control; *Fusarium equiseti*; biological characteristics; antifungal activity

## 1. Introduction

*Rehmannia glutinosa* Libosch is a perennial herbaceous plant cultivated in Asia, including China [1]. The root tuber of *R. glutinosa* is rich in phenylethanol glycosides, iridoid glycosides, polysaccharides, and other chemical components, and it has various pharmacological effects on the cardiovascular, cerebrovascular, central nervous, immune, and visceral systems [2–4]. *R. glutinosa* has a long history of cultivation in many China provinces, including Henan, Shandong, Shanxi, and Shaanxi [5]. *R. glutinosa* cultivated in Wen County, Henan Province is known as one of the four famous "Huai" medicines, which is famous worldwide for its high efficacy [6].

Replanted disease seriously limits the production and cultivation of *R. glutinosa*. Replanted *R. glutinosa* often suffers from disease, and this decreases the yield, quality, and accumulation of active ingredients [7]. *Rhizoctonia solani* and *Fusarium proliferatum* may cause root rot in *R. glutinosa*, which can induce the rotting of both the stem and roots and eventually result in the widespread death of *R. glutinosa* in fields [8,9]. *Phoma herbarum* and *Ascochyta molleriana* may cause ring rot in *R. glutinosa*, which is observed more frequently on the bottom leaves, and this can lead to the formation of large disease spots [10]. *Septoria digitalis* may cause spot blight in *R. glutinosa*. The spots associated with spot blight are round or irregular, and many black conidia are scattered around the spots [11]. *Plectosphaerella*

*cucumerina* may infect entire *R. glutinosa* plants and induce wilting; plants will rot and die when the disease is serious [12]. *Alternaria alternata* may cause brown spot in *R. glutinosa*, which is characterized by the formation of round or irregular brown spots on the leaves, with yellow halos at the edges of the spots [13].

Few studies have examined strategies to control diseases of *R. glutinosa*. The main approach used to control diseases of *R. glutinosa* is to reduce the nutrient sources of pathogenic fungi and the occurrence of diseases through crop rotations with wheat and corn. Only three fungicides registered in China can be used for the control of *R. glutinosa* diseases: metalaxyl-M·hymexazol 32% solution (for treating seeds), fluazinam·iprodione 40% aqueous suspension concentrate, and *Bacillus subtilis* 1 billion spores/g wettable powder. These three fungicides have been used to control root rot in *R. glutinosa*; however, there are no registered fungicides for leaf spot disease (http://www.chinapesticide.org.cn/hysj/index.jhtml, accessed on 21 January 2023). Most farmers use fungicides according to their own experience, yet this is often not effective for controlling the various diseases of *R. glutinosa* [8]. In 2019, outbreaks of leaf spot disease occurred in *R. glutinosa* fields in Henan Province, China. In this study, a new pathogen causing leaf spot disease in *R. glutinosa* was isolated and identified. The biological characteristics of the new isolated pathogenic fungus were studied. To screen fungicides that could be used to control the leaf disease caused by *Fusarium* spp. in *R. glutinosa*, the activities of eight fungicides against the new pathogen and *F. acuminatum* causing the leaf blight of *R. glutinosa* were determined.

## 2. Materials and Methods

### 2.1. Sampling and Isolation

In July 2019, *R. glutinosa* plants with typical leaf disease symptoms were collected from the *R. glutinosa* fields in Wen County (35°03′31″N, 113°07′17″E), Wuzhi (36°09′21″N, 114°71′47″E), and Yuanyang (35°07′00″N, 113°57′23″E), Henan Province, China.

Pathogens were isolated and purified following the procedures of Guo [14], with minor modifications. Briefly, infected leaf samples were cleaned with sterile water 3 times and then dried with sterile absorbent paper. Pieces of infected leaf tissues (5 × 5 mm) were carefully removed from lesion junctures using sterile scalpels. Next, infected leaf sample tissues were surface-sterilized with 75% ethanol for 30 s and 2% sodium hypochlorite for 90 s, rinsed three times with sterilized distilled water, and incubated on PDA containing 50 µg·mL$^{-1}$ cephalosporin. The plates were incubated at 25 °C with a 12-h photoperiod. After mycelia grew, single-root hyphal tip growth points from the colony edges were picked under a stereomicroscope for cultivation to obtain purified isolates.

### 2.2. Pathogenicity Determination

Healthy *R. glutinosa* plants (5–8 leaves) were used to test the pathogenicity of each isolate. Mycelial plugs (4 mm in diameter) from 4-day-old cultures grown on PDA were placed on the leaf surfaces of 15 *R. glutinosa* plants that had been injured by a sterile needle. Three healthy leaves on each plant were selected, and one mycelial plug was placed on each leaf. PDA-only plugs placed on the leaf surfaces of 15 plants served as controls. The plants were placed in a growth chamber at 25 °C with 80% to 85% relative humidity under a 12-h photoperiod for 6 to 8 days [15]. Disease symptoms on the leaves were observed daily after inoculation with mycelial plugs. Subsequently, the diseased leaves were collected, and the pathogenic fungi were re-isolated and identified to fulfil Koch's postulates.

### 2.3. Morphological Observations of Pathogens

For the morphological observations, a mycelial plug with a 4 mm diameter was taken from the edge of each 4-day-old colony and cultured on PDA at 25 °C. The morphology and growth of the fungal colonies were recorded regularly every day. The conidiophores and conidia of the pathogen were observed, and the sizes were measured under an inverted fluorescence microscope (Nikon Eclipse Ti-S) after each colony had been cultured on PDA for 15 days.

*2.4. DNA Extraction and Multiple Sequence Analysis of Pathogens*

The fungal genomic DNA was extracted using the cetyltrimethylammonium bromide method [16] from mycelia cultured on PDA for 5 days. The internal transcribed spacer (ITS) sequence was amplified using the primers ITS1 (5′-TCCGTAGGTGAACCTGCGG-3′) and ITS4 (5′-TCCTCCGCTTATTGATATGC-3′) [17] from the total genomic DNA of each isolate. The primers TUB2T1 (5′-AACATGCGTGAGATTGTAAGT-3′) and Bt2b (5′-ACCCTCAGTGTAGTGACCCTTGGC-3′) [18,19] were used to amplify the β-tubulin (*Tub*) sequence. The primers EF-728F (5′-CATCGAGAAGTTCGAGAAGG-3′) and EF-986R (5′-TACTTGAAGGAACCCTTACC-3′) [20] were used to amplify the elongation factor 1-α (*EF1-α*) sequence.

The PCR reaction mixture contained 22 μL of 1.1 × T3 Super PCR Mix (Tsingke Biotechnology Co., Ltd., Beijing, China), 1 μL of each primer, and 1 μL of DNA template. The ITS region's amplification reaction program was as follows: pre-denaturation at 98 °C for 2 min; 35 cycles of denaturation at 98 °C for 10 s, annealing at 55 °C for 10 s, and extension at 72 °C for 10 s; and a final extension at 72 °C for 2 min. The annealing temperatures of the *EF1-α* and *Tub* genes were 55 °C and 53 °C, respectively, and the reaction steps and other temperatures were the same as those for ITS amplification. The amplified products were detected using 1% agarose gel electrophoresis and sequenced. DNA sequence homology searches were performed using online BLAST searches of GenBank, and the corresponding ITS regions, *Tub*, and *EF1-α* sequences were downloaded (Table 1). A multiple sequence alignment was performed using ClustalW [21]. The aligned sequences were concatenated using the Concatenate Sequence tool after the trimAI modification [22]. A phylogenetic tree was constructed using the neighbor-joining method in MEGA 7.0 [23]. *Verticillium dahliae* was used as the outgroup, and branch support was evaluated using 1000 bootstrap replicates.

**Table 1.** Isolates used for phylogenetic analyses in the study and their GenBank accession numbers.

| Species | Strain | GenBank Accession Numbers | | | Reference |
|---|---|---|---|---|---|
| | | ITS | *Tub* | *EF1-α* | |
| *F. acuminatum* | IBE000006 | EF531232 | EF531244 | EF531237 | [24] |
| *F. acuminatum* | LD1508081502 | MF523230 | MF523225 | MF523228 | NP |
| *F. acuminatum* | HHAUF210502 | MZ351204 | MZ366376 | MZ366377 | ● |
| *F. acuminatum* | HHAUF210515 | OP846525 | OP837541 | OP837544 | ● |
| *F. ambrosium* | CBS 571.94 | KM231801 | KM232063 | KM231929 | [25] |
| *F. avenaceum* | M127 | KP265352 | KP710634 | KP674195 | [26] |
| *F. equiseti* | HHAUF210507 | MZ543968 | MZ547046 | MZ547045 | ● |
| *F. equiseti* | HHAUF210509 | OP846523 | OP837539 | OP837542 | ● |
| *F. equiseti* | HHAUF210513 | OP846524 | OP837540 | OP837543 | ● |
| *F. equiseti* | HGUP17361.1 | MK069606 | MK069604 | MK069605 | NP |
| *F. equiseti* | MAFF 236434 | AB586999 | AB587047 | AB674277 | [27] |
| *F. equiseti* | UP-PA002 | MH521295 | MH521296 | MH521297 | [28] |
| *F. graminearum* | M216A | KP295509 | KP765707 | KP400687 | [26] |
| *F. incarnatum* | MAFF 236386 | AB820720 | AB820712 | AB820704 | [27] |
| *F. incarnatum* | MAFF 236521 | AB586988 | AB587036 | AB674267 | [29] |
| *F. oxysporum* | M228 | KP264651 | KP674278 | KP400703 | [26] |
| *F. poae* | FRCT-0796 | AB586983 | AB587072 | AB674301 | [29] |
| *F. redolens* | Z318 | KP264660 | KP674235 | KP400713 | [26] |
| *F. sambucinum* | M120 | KP265348 | KP710630 | KP674191 | [26] |
| *F. solani* | Z141 | KP265362 | KP710644 | KP674205 | [26] |
| *F. tricinctum* | SPF001 | MG990937 | MG990938 | MG990939 | [30] |
| *F. tricinctum* | SPF003 | MG704912 | MG704913 | MG704914 | [30] |
| *F. tricinctum* | MAFF 235.551 | AB587030 | AB587079 | AB674262 | [31] |
| *F. verticillioides* | CBS 114579 | KU604025 | KU603860 | KU711696 | NP |
| *Verticillium dahliae* | Vd8 | HE972024 | KF555285 | KM408514 | [32] |

●: GenBank accession numbers of HHAUF210502, HHAUF210507, HHAUF210509, HHAUF2105013, and HHAUF210515. NP: unpublished.

### 2.5. Determination of the Optimal Growth Conditions of the Pathogen

The mycelial plugs used in this study were all cut from the margins of the pathogenic fungal colonies that had grown for 4 days on PDA and achieved diameters of 4 mm. Each isolate was incubated at a constant temperature for 4 days and the experiment was conducted using three replicates. The diameters of the colonies were measured using the cross method.

To characterize the effects of different media on mycelial growth, the mycelial plugs were inoculated onto PDA, PSA, PCA, OA, CMA, CZ, and WA medium [12] (Table 2) and cultured at 25 °C. To clarify the effect of light on mycelial growth, the mycelial plugs were inoculated onto the center of the PDA, and the illumination conditions included all light, half-light, and all dark. The colonies were cultured at 25 °C. To investigate the effect of pH on mycelial growth, 1 mol L$^{-1}$ HCl and NaOH were used to adjust the pH of the PDA to 4, 5, 6, 7, 8, 9, 10, and 11. Mycelial plugs were then inoculated on the PDA at various pH levels and cultured at 25 °C. To clarify the effect of carbon and nitrogen sources on mycelial growth, CZ medium with glucose, maltose, fructose, sucrose, lactose, and soluble starch as the sole carbon source was used. Media with different nitrogen sources were made using equal amounts of sodium nitrate, potassium nitrate, protein, glycine, cysteine, ammonium chloride, ammonium sulfate, and ammonium nitrate as the sole nitrogen source. The mycelial plugs were inoculated on the center of the CZ and cultured at 25 °C. To clarify the effect of temperature on mycelial growth, mycelial plugs were inoculated on the center of the PDA and cultured in constant temperature incubators set at 5 °C, 10 °C, 15 °C, 20 °C, 25 °C, 30 °C, 35 °C, and 40 °C. To determine the lethal mycelial temperature, mycelial plugs were placed in water baths set at 40 °C, 43 °C, 46 °C, 49 °C, 52 °C, 55 °C, 58 °C, and 61 °C for 10 min. They were then transferred to the PDA for constant temperature cultivation at 25 °C, and colony growth was observed and recorded daily. After the approximate lethal mycelial temperature was determined, the temperature gradient interval was reduced to 1 °C to more precisely determine the lethal temperature.

**Table 2.** Formulation of the media.

| Medium | Formulation |
| --- | --- |
| PDA | peeled potato, 200 g; glucose, 20 g; agar powder, 20 g, distilled water to 1 L |
| PSA | peeled potato, 200 g; sucrose, 20 g; agar powder, 20 g, distilled water to 1 L |
| PCA | peeled potato, 20 g; peeled carrot, 20 g; agar powder, 20 g, distilled water to 1 L |
| OA | oatmeal, 30 g; agar powder, 20 g, distilled water to 1 L |
| CMA | corn flour, 50 g; agar powder, 15 g, distilled water to 1 L |
| CZ | NaNO$_3$, 3 g; K$_2$HPO$_4$•3H$_2$O, 1 g; MgSO$_4$•7H$_2$O, 0.5 g; KCl, 0.5 g; FeSO$_4$•7H$_2$O, 0.01 g; sucrose, 30 g; agar powder, 20 g, distilled water to 1 L |
| WA | agar powder, 20 g, distilled water to 1 L |

### 2.6. Bioassays

The inhibitory activities of eight fungicides against pathogens were determined using the mycelial growth rate method [33,34]. Thiophanate-methyl was dissolved in dimethyl sulfoxide, and the solution was diluted to five concentrations (Table 3) containing 1% dimethyl sulfoxide. Boscalid (250 mg), chlorothalonil (250 mg), and five other fungicides (25 mg each) were dissolved in 1 mL of acetone and then diluted into five concentrations containing 2% acetone solution. The diluted solution and PDA were mixed at a 1:9 volume ratio and then poured into 9 cm Petri dishes. After solidification, a 4 mm mycelial plug was cut from the edge of a 4-day-old colony and reverse-inoculated into the center of the Petri dish. Petri dishes without fungicides (containing 1% dimethyl sulfoxide or 2% acetone) were used as experimental controls. Each treatment was repeated in triplicate. The dishes

were then incubated at 25 °C for 4 days. Colony diameters on each plate were measured in two perpendicular directions, and the diameter of the original mycelial plug (4 mm) was subtracted from this measurement. The growth inhibition rates were calculated using the following formula:

% Inhibition rate of mycelial growth = 100 × (Colony diameter of control group − colony diameter of experimental group)/(Colony diameter of control group-4 mm).

**Table 3.** Fungicides and experimental concentration gradients.

| Fungicides | Manufacturer | Concentration Gradient (mg·L$^{-1}$) |
|---|---|---|
| Thiophanate-methyl (97% a.i.) | Anhui Guangxin Agrochemical Co., Ltd. | 5, 10, 20, 40, 80, 160 |
| Boscalid (97% a.i.) | Hebei Lansheng Biotechnology Co., Ltd. | 12.5, 25, 50, 100, 200, 400 |
| Prochloraz (97% a.i.) | Jiangxi Huihe Chemical Co., Ltd. | 0.25, 0.5, 1, 2, 4, 8 |
| Flusilazole (97% a.i.) | Shandong Audley Chemical Co., Ltd. | 0.125, 0.25, 0.5, 1, 2, 4 |
| Tebuconazole (97% a.i.) | Xinyi Youlian Chemical Co., Ltd. | 0.25, 0.5, 1, 2, 4, 8 |
| Trifloxystrobin (97% a.i.) | Nanjing Kangmanlin Chemical Co., Ltd. | 1, 2, 4, 8, 16, 32 |
| Pyraclostrobin (98% a.i.) | Hebei Chengyue Chemical Co., Ltd. | 0.25, 0.5, 1, 2, 4, 8 |
| Chlorothalonil (98% a.i.) | Anhui Guangxin Agrochemical Co., Ltd. | 12.5, 25, 50, 100, 200, 400 |

*2.7. Data Analyses*

The logarithmic value of each concentration of each test agent was used as the independent variable (X), and the probability value of the mycelial growth inhibition rate was used as the dependent variable (Y) to establish the virulence regression equation. Statistical analyses were conducted using SPSS 23.0 software (SPSS Inc., IBM Corp., Armonk, NY, USA) for Windows. The median effective concentrations (EC$_{50}$) were determined by regression analyses of the inhibition rates against the log10 concentrations of the fungicides. A one-way analysis of variance was used to analyze the differences among variables, including EC$_{50}$ values, obtained under different culture conditions, such as culture medium, temperature, pH, light, and carbon and nitrogen sources.

## 3. Results and Analyses

### 3.1. Pathogen Isolation and Pathogenicity Determination

Twenty-five isolates were obtained from the diseased leaf samples (Figure S1). The pathogenicity tests showed that *R. glutinosa* could be infected by five isolates, including HHAUF210502 and HHAUF210507 isolated from Wenxian, HHAUF210515 and HHAUF210 509 isolated from Wuzhi, and HHAUF210513 isolated from Yuanyang. Previously, we reported that leaf spot of *R. glutinosa* can be induced by *Fusarium acuminatum* HHAUF210502 [35].

Leaves of *R. glutinosa* inoculated with HHAUF210507, HHAUF210509, and HHAUF210 513 showed similar symptoms. After 2 days, they developed dark brown spots at the inoculation site. The spots were surrounded by leaf veins and formed irregular polygons, and developed clear boundaries. Small amounts of white mycelia emerged on the backs of the leaf lesions. After 7 days, the spots gradually expanded and turned tawny, and the leaves withered and wrinkled at the spots (Figure 1A–C). Two days after *R. glutinosa* was inoculated with HHAUF210502 and HHAUF210515, there were round or irregular plaques at the inoculation site, which were dark brown, rotten, and perforated, and these were accompanied by pink mycelia. After 7 days, the leaves became green and shrunk (Figure 1D,E). These lesions were similar to those observed on leaves in the field (Figure S2), and the control leaves remained symptomless. The fungi were reisolated from the diseased leaves, thereby satisfying Koch's postulates.

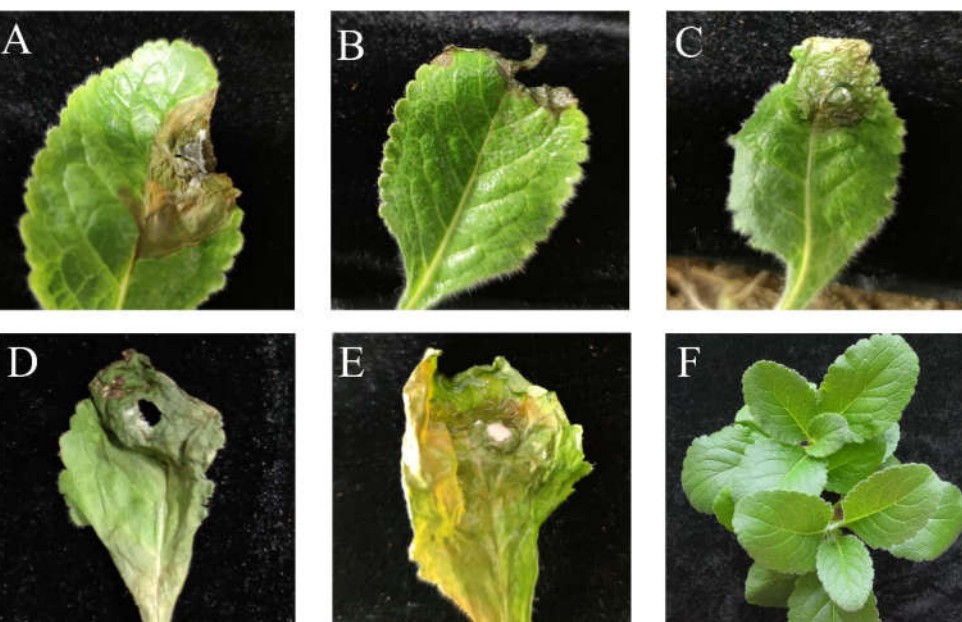

**Figure 1.** Symptoms of *Rehmannia glutinosa* leaves inoculated with pathogens at 7 days. (**A**) Leaves inoculated with HHAUF210507; (**B**) leaves inoculated with HHAUF210509; (**C**) leaves inoculated with HHAUF2105013; (**D**) leaves inoculated with HHAUF210502; (**E**) leaves inoculated with HHAUF2105015; (**F**) control.

### 3.2. Morphological Characterization of the Pathogens

The colony morphology of HHAUF210507, HHAUF210509, and HHAUF210513 was similar. The pathogens developed aerial mycelia, which were white. The colonies were white at the beginning and gradually became light camel-colored over time, and the mycelium of HHAUF210513 was sparse compared with the other two isolates (Figure 2(A1–A3)). Macroconidia (17.4–46.2 × 2.3–4.2 μm) were sickle-shaped, curved at both ends, enlarged in the middle, and had three to five septa (Figure 2(B1–B3)). There were few elliptical microconidia (7.5–11.8 × 1.8–2.9 μm) with zero to one septa (Figure 2(B1–B3)). The chlamydospores (4.8–8.2 μm) were oblate and were arranged in strings among the mycelia (Figure 2(C1–C3)). The typical phialids–conidiophores arose from the mycelia (Figure 2(D1–D3)). HHAUF2105 07, HHAUF210509, and HHAUF210513 were preliminarily identified as *F. equiseti* on the basis of morphological characteristics.

Both HHAUF210502 and HHAUF210515 developed abundant aerial mycelia that were fleece-shaped; the mycelia were white and had a slight bulge in the center of the colony. The colonies were initially red. The center of the colony gradually became dark brown, and the margin became dark brown to earth yellow (Figure 2(A4,A5)). Macroconidia (17.3–24.5 × 3.6–5.8 μm) were sickle-shaped, slightly pointed at both ends, and had three to eight septa (Figure 2(B4,B5)). Microconidia (5.6–10.7 × 3.0–4.4 μm) were ovoid and had zero to one septa (Figure 2(B4,B5)). The chlamydospores (7.2–13.6 μm) were oblate and were arranged in strings among the mycelia (Figure 2(C4,C5)). The typical phialids–conidiophores arose from the mycelia (Figure 2(D4,D5)). HHAUF210502 and HHAUF210515 were preliminarily identified as *F. acuminatum* on the basis of morphological characteristics.

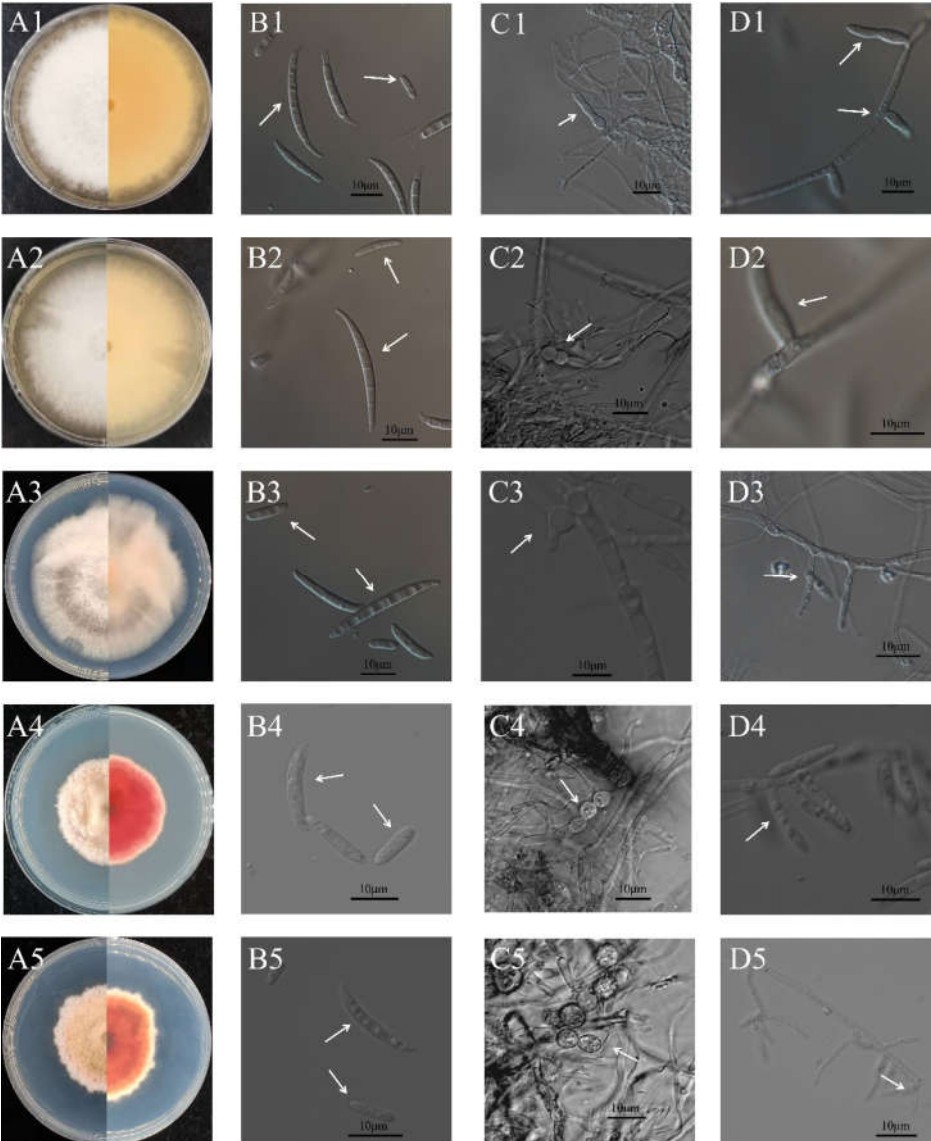

**Figure 2.** Morphological characteristics of pathogens. (**A**) The front (left) and back (right) of a colony; (**B**) macroconidia and microconidia; (**C**) chlamydospores; (**D**) conidiophores. B, C and D are indicated by arrows. Numbers 1, 2, 3, 4, and 5 correspond to HHAUF210507, HHAUF210509, HHAUF210513, HHAUF210502, and HHAUF210515, respectively.

*3.3. Phylogenetic Analysis of Pathogens*

To identify these pathogens, their total genomic DNA was used as a template for PCR amplification and sequencing. We obtained 687 bp (ITS, MZ543968), 1479 bp (*Tub*, MZ547046), and 576 bp (*EF1-α*, MZ547045) fragments for HHAUF210507; 677 bp (ITS, OP846523), 962 bp (*Tub*, OP837539), and 562 bp (*EF1-α*, OP837542) fragments for HHAUF210 509; 684 bp (ITS, OP846524), 962 bp (*Tub*, OP837540), and 562 bp (*EF1-α*, OP837543) fragments for HHAUF210513; 567 bp (ITS, MZ351204), 960 bp (*Tub*, MZ366376), and 290 bp (*EF1-α*, MZ366377) fragments for HHAUF210502; and 566 bp (ITS, OP846525), 960 bp (*Tub*, OP837541), and 290 bp (*EF1-α*, OP837544) fragments for HHAUF210515. A BLAST analysis showed that the similarity of fragments belonging to HHAUF210507, HHAUF210509, and HHAUF210513 with previously submitted *F. equiseti* sequences (KJ396338, MK334366, and KP732017) was 99% to 100%, and the similarity of the fragments belonging to HHAUF210502 and HHAUF210515 with previously submitted *F. acuminatum* sequences (KU852603, OM956 058, and MH822061) was 99% to 100%. A phylogenetic tree was constructed using

ITS (398 bp), *Tub* (295 bp), and *EF1-α* (220 bp) sequences (Figure 3). HHAUF210507, HHAUF210509, and HHAUF210513 were clustered in the same evolutionary clade with *F. equiseti* UP-PA002, HGUP17361.1, and MAFF 236434, with 99% bootstrap support. Thus, on the basis of the morphological and molecular characteristics, HHAUF210507, HHAUF210509, and HHAUF210513 isolated from diseased leaves of *R. glutinosa* in Wen County, Henan Province, China, were identified as *F. equiseti*. HHAUF210502 and HHAUF21 0515 were clustered in the same evolutionary clade with *F. acuminatum* IBE000006 and LD1508081502, with 93% bootstrap support. Thus, on the basis of the morphological and molecular characteristics, HHAUF210502 and HHAUF210515 isolated from diseased leaves of *R. glutinosa* in Wen County, Henan Province, China, were identified as *F. acuminatum*.

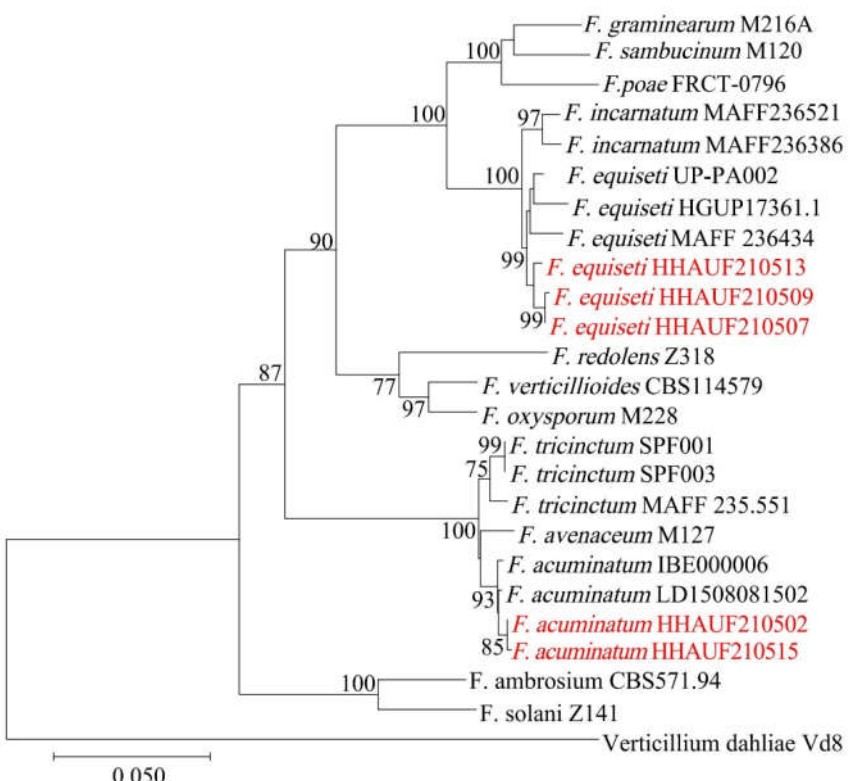

**Figure 3.** Phylogenetic tree inferred from concatenated ITS, *Tub*, and *EF1-α* sequences using the neighbor-joining method. Bootstrap values (%) presented at the branches were calculated from 1000 replications. Values below 50% are hidden. The scale bar indicates a 5% sequence difference.

### 3.4. The Optimal Growth Conditions of F. equiseti

The mycelial colonies of *F. equiseti* isolated from *R. glutinosa* completely covered the medium in each plate after 5 days under different light treatments. The growth rate was fastest when the light:dark ratio was 1:1, with the colony diameter reaching 7.19 cm after 4 days. This was significantly larger than the diameters achieved under all light or all dark conditions (Figure 4A). The isolate was not sensitive to alkalinity, and grew normally even under strong alkaline conditions. The optimal pH for mycelial growth was 10.0, and the average colony diameter was 6.85 cm after 4 days. Under acidic conditions, the mycelial growth rate slowed as the pH decreased, and mycelia did not grow at a pH of 3 (Figure 4B). The isolate grew between 5 °C and 35 °C, and the optimal growth temperature was 25 °C. The colony diameter reached 6.53 cm at 25 °C after 4 days, and this was significantly larger than diameters achieved at other temperatures (Figure 4C). Mycelia stopped growing at 40 °C, and the lethal mycelial temperature was 55 °C. The isolate could grow on PDA, PSA, PCA, OA, CMA, CZ, and WA medium. The growth rate on CZ was the fastest, and the whole plate was covered by a white colony after 5 days. The colony diameter was 6.86 cm after 4 days. This was significantly larger than diameters achieved on other tested media

(Figure 4D). However, the mycelia on CZ were sparse and in poor condition. The colonies cultured on PSA were pale orange-red, and the mycelia were dense. The average colony diameter (6.06 cm) was smaller on PSA than on CZ, PDA, and PCA. The colony diameter on PDA was 6.53 cm after 4 days, and the mycelia were white and dense. PDA might be the most suitable medium for the growth of *F. equiseti*. When soluble starch was used as the carbon source, the mycelia were dense and grew the fastest, and the average colony diameter was 7.31 cm. The utilization rate of fructose by the isolate was the lowest, and the average colony diameter was only 5.78 cm (Figure 4E). When peptone was used as the nitrogen source, the mycelia were the densest and grew the fastest, and the average colony diameter was 7.54 cm after 4 days. When cysteine was used as the nitrogen source, mycelia grew the slowest, and the average colony diameter was only 4.69 cm after 4 days, which was significantly smaller than colonies observed when other nitrogen sources were used (Figure 4F).

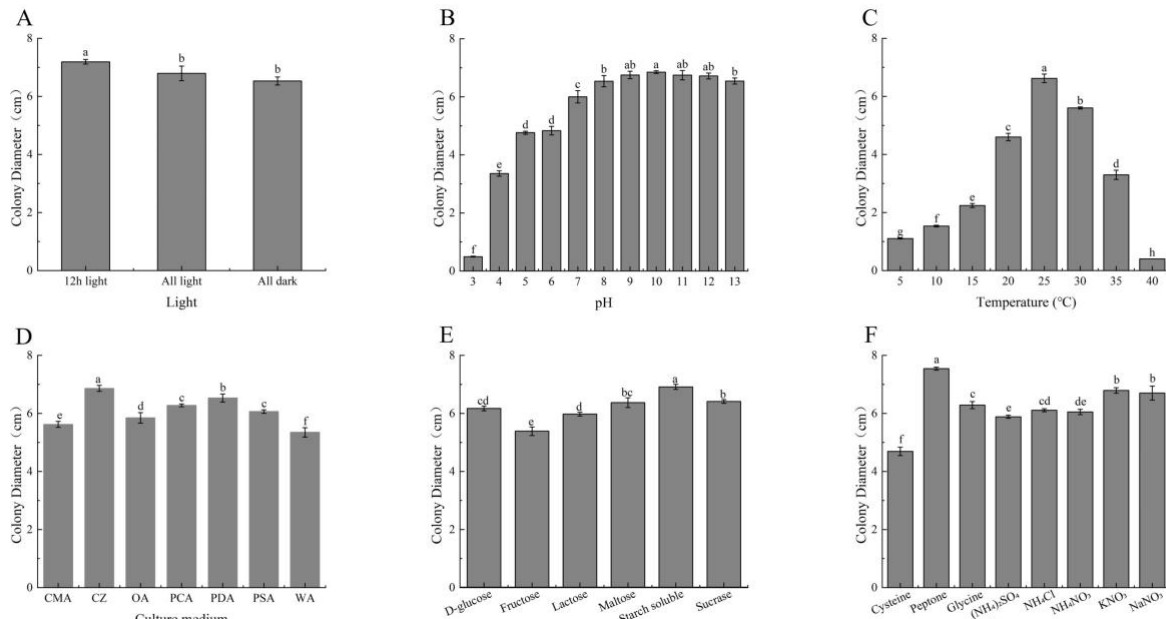

**Figure 4.** Biological characteristics of *F. equiseti*. (**A**) Light condition; (**B**) pH; (**C**) temperature; (**D**) medium; (**E**) carbon source; (**F**) nitrogen source. Different letters indicate significant differences according to ANOVA.

*3.5. Activities of Fungicides against F. equiseti and F. acuminatum*

Eight fungicides conventionally used in agricultural fields were screened for their antifungal activities against *F. equiseti* and *F. acuminatum*, and all eight of the tested agents had inhibitory activities against both pathogens (Tables 4 and 5). Among the eight fungicides, prochloraz was most effective in suppressing the mycelial growth of *F. equiseti* (EC$_{50}$ of 0.139 mg·L$^{-1}$) and *F. acuminatum* (EC$_{50}$ of 0.123 mg·L$^{-1}$). This was followed by tebuconazole and flusilazole, with EC$_{50}$ values of 0.194 and 0.318 mg·L$^{-1}$ for *F. equiseti*, respectively, and with EC$_{50}$ values of 0.617 and 0.688 mg·L$^{-1}$ for *F. acuminatum*, respectively. Pyraclostrobin also had strong inhibitory activity, with EC$_{50}$ values of 4.671 and 2.22 mg·L$^{-1}$ for *F. equiseti* and *F. acuminatum*, respectively. The antifungal activities of trifloxystrobin and thiophanate-methyl were lower, with EC$_{50}$ values of 25.158 and 35.261 mg·L$^{-1}$ for *F. equiseti*, respectively, and with EC$_{50}$ values of 21.505 and 37.249 mg·L$^{-1}$ for *F. acuminatum*, respectively. The antifungal activities of boscalid and chlorothalonil were the lowest, with EC$_{50}$ values of 204.807 and 215.96 mg·L$^{-1}$ for *F. equiseti*, respectively, and with EC$_{50}$ values of 173.158 and 147.666 mg·L$^{-1}$ for *F. acuminatum*, respectively.

**Table 4.** Antifungal activities of fungicides against *F. equiseti*.

| Fungicides | Regression Equation | EC$_{50}$ (mg·L$^{-1}$) | Correlation Coefficient | 95% Confidence Intervals (mg·L$^{-1}$) |
|---|---|---|---|---|
| Thiophanate-methyl (97% a.i.) | Y = 2.704 + 1.484X | 35.261 | 0.970 | 29.552~42.074 |
| Boscalid (97% a.i.) | Y = 3.637 + 0.590X | 204.807 | 0.990 | 118.740~353.260 |
| Prochloraz (97% a.i.) | Y = 6.140 + 1.329X | 0.139 | 0.986 | 0.082~0.234 |
| Flusilazole (97% a.i.) | Y = 5.769 + 1.548X | 0.318 | 0.980 | 0.257~0.395 |
| Tebuconazole (97% a.i.) | Y = 5.817 + 1.146X | 0.194 | 0.991 | 0.118~0.317 |
| Trifloxystrobin (97% a.i.) | Y = 4.548 + 0.323X | 25.158 | 0.963 | 7.797~81.173 |
| Pyraclostrobin (98% a.i.) | Y = 4.711 + 0.432X | 4.668 | 0.995 | 2.175~10.020 |
| Chlorothalonil (98% a.i.) | Y = 3.266 + 0.743X | 215.960 | 0.987 | 163.783~340.970 |

**Table 5.** Antifungal activities of fungicides against *F. acuminatum*.

| Fungicides | Regression Equation | EC$_{50}$ (mg·L$^{-1}$) | Correlation Coefficient | 95% Confidence Intervals (mg·L$^{-1}$) |
|---|---|---|---|---|
| Thiophanate-methyl (97% a.i.) | Y = 2.582 + 1.539X | 37.249 | 0.974 | 19.764~70.204 |
| Boscalid (97% a.i.) | Y = 2.317 + 1.199X | 173.158 | 0.994 | 64.197~467.060 |
| Prochloraz (97% a.i.) | Y = 5.606 + 0.667X | 0.123 | 0.980 | 0.050~3.313 |
| Flusilazole (97% a.i.) | Y = 5.186 + 1.142X | 0.688 | 0.987 | 0.313~1.511 |
| Tebuconazole (97% a.i.) | Y = 5.237 + 1.131X | 0.617 | 0.995 | 0.225~1.690 |
| Trifloxystrobin (97% a.i.) | Y = 4.968 + 0.774X | 21.505 | 0.983 | 3.641~127.012 |
| Pyraclostrobin (98% a.i.) | Y = 4.723 + 0.799X | 2.223 | 0.987 | 0.686~7.203 |
| Chlorothalonil (98% a.i.) | Y = 3.298 + 0.785X | 147.666 | 0.966 | 39.031~558.666 |

## 4. Discussion

*F. equiseti* is an important pathogen worldwide; it infects wheat, barley, maize, rice, and other crops and causes diseases, such as root rot, stem rot, and head blight [36–38]. It also infects vegetable and fruit crops and causes diseases such as chili wilt [39], coriander stem and root rot [40], postharvest *Fusarium* rot of mandarin [41], and leaf spots on mango [42]. Some trees, such as *Aleppo pine* [43] and *Avicennia marina* [44], can also be infected by *F. equiseti*. Chinese medicinal herbs are damaged by *F. equiseti* infections. For example, dieback disease on *Dendrobium officinale* [14], discoloration of *Panax quinquefolius* roots [45], and shoot blight of *Atractylode schinensis* [46] are serious diseases. This study reports for the first time the occurrence of leaf spot on *R. glutinosa* caused by *F. equiseti.* The symptoms of leaf spot were described, and the pathogen was identified using morphological and molecular characteristics.

Disease in *R. glutinosa* caused by *F. equiseti* was mainly observed in late July and August. During the early stage of infection, dark brown spots with distinct boundaries appeared at the leaf tips. As the disease developed, the disease spots gradually expanded into irregular polygons and were surrounded by veins. Parts of the disease spots produced small amounts of white aerial mycelia. The disease spots gradually spread along the veins. The leaves withered and wrinkled, and finally the aboveground part of *R. glutinosa* completely died. This is similar to the leaf disease caused by *F. acuminatum*; however, the lesions caused by *F. acuminatum* do not show obvious boundaries, and the lesions spread faster. Members of the *Fusarium* genus have often been blamed for the root rot of *R. glutinosa* [9]; however, when treated with *F. equiseti* spore suspensions, the roots of *R. glutinosa* grew healthily without rotting. This characteristic of leaf-specific infections suggests that foliar spraying could control *R. glutinosa* diseases caused by *F. equiseti*. In this study, *F. equiseti* isolated from *R. glutinosa* leaves only caused leaf spot disease, which appears to represent a special case. Therefore, the biological characteristics of the isolate were further studied.

*F. equiseti* isolated from *R. glutinosa* grew fastest at 25 °C and pH of 9–12, with a 12 h light/12 h dark photoperiod. The highest utilization rates were achieved with soluble

starch and peptone. The optimal medium for the growth of this isolate might be PDA. Significant differences were observed in the lethal mycelial temperature (55 °C), optimal growth temperature, and carbon source between *F. acuminatum* and *F. equiseti* isolated from the leaves of *R. glutinosa*. The light-related characteristics of *F. equiseti* isolated in this study differed from those of *F. equiseti*, which causes shoot blight of *A. schinensis* [46] and the isolate causing maize ear rot [47]. The isolate that causes shoot blight of *A. schinensis* grows vigorously in full light, whereas the isolate that causes maize ear rot grows in total darkness. This difference may be related to the differences in the plant organs targeted for infection. In addition, the optimum pH for the growth of *F. equiseti* isolated in this study was similar to that of the isolate causing shoot blight of *A. schinensis*, with both growing vigorously at a pH of 8 to 12. However, the isolate causing maize ear rot grows vigorously at a lower pH (6 to 7). These differences may be related to the species, geographical distribution, or growth habits of the host. *F. equiseti* may have a wide host range. Therefore, the optimal strategy for controlling the spread of *F. equiseti* depends on the isolate. These three isolates had the same optimal growth temperature range of 25–30 °C, which is consistent with the temperature of the luxuriant leaf period and root enlargement period of *R. glutinosa* from July to August [9]. The number of leaves on *R. glutinosa* increases rapidly in July and August, which provides favorable conditions for pathogen infection. Thus, water and fertilizer management in early July requires attention when developing strategies to prevent and control disease. *F. equiseti* isolated in this study was highly adaptable to alkaline conditions, and its growth rate declined as the pH decreased, indicating that acidic conditions may inhibit the leaf spot of *R. glutinosa* caused by *F. equiseti*. The pH of the medium after colonization by *F. equiseti* isolated in this study was detected, and the isolate significantly reduced the pH of the medium. It was then maintained at pH 8, suggesting that *F. equiseti* isolated in this study has the ability to reduce soil alkalinity.

The bioassays showed that the selected eight fungicides had antifungal effects against *F. equiseti* and *F. acuminatum*, and the same fungicide had similar antifungal levels against the two *Fusarium* species; these data indicate that these fungicides could be used for the control of most *Fusarium* species. Triazole fungicides are effective for controlling *Fusarium* spp. [48,49]. Flusilazole and tebuconazole selected in this study have high antifungal activities against *F. equiseti* and *F. acuminatum*, which is consistent with the results of a previous study [48]. In addition, Mengesha et al. [50] reported that tebuconazole is effective for controlling wheat scab; thus, triazole fungicides such as flusilazole and tebuconazole could be used to control leaf spot of *R. glutinosa* caused by *Fusarium* spp. In our study, prochloraz showed high antifungal activity against *F. equiseti* and *F. acuminatum*. Prochloraz is commonly used to control plant leaf diseases caused by *Fusarium* spp. [51]. Mateo et al. [52] found that prochloraz had a stronger inhibitory effect on the growth of *F. langsethiae* mycelia than tebuconazole, which is similar to the results of our study. Therefore, prochloraz could also be used as a fungicide to control the leaf spot of *R. glutinosa* caused by *Fusarium* spp. Malandrakis et al. [53] determined the antifungal activity of nine fungicides against *F. solani*; the $EC_{50}$ of pyraclostrobin was 5 mg·L$^{-1}$, and the $EC_{50}$ of boscalid and thiophanate-methyl was more than 100 mg·L$^{-1}$, which is similar to the results of our study. Liao et al. [54] suggested that thiophanate-methyl might effectively inhibit the mycelial growth of *F. solani*, the pathogen of roselle root rot, whereas Ghosal et al. [55] found that thiophanate-methyl weakly inhibited the spore germination of *F. oxysporum*. These contrasting results might be related to the species of pathogen, indicating that thiophanate-methyl is not suitable for the control of all *Fusarium* spp. Few studies have examined the antifungal activity of chlorothalonil against *Fusarium* spp. The results of our study indicate that chlorothalonil had a weak inhibitory effect on *F. equiseti* and *F. acuminatum* compared with other fungicides; thus, it should not be used as a control agent. Given that the results of laboratory bioassays do not always provide an accurate reflection of control efficacies in the field, field trials must be conducted.

## 5. Conclusions

In sum, five isolates of pathogenic fungi were isolated from diseased leaves of *R. glutinosa* plants, and a new pathogen causing leaf disease of *R. glutinosa* was identified as *F. equiseti*. The biological characteristics of *F. equiseti* were clarified, and control agents against *F. equiseti* and *F. acuminatum* were screened. Among the eight fungicides, prochloraz had the highest antifungal activity against *F. equiseti* and *F. acuminatum*. Our data expand our understanding of leaf diseases in *R. glutinosa* and will promote the development of management strategies for controlling leaf diseases of *R. glutinosa* caused by *F. equiseti*.

**Supplementary Materials:** The following supporting information can be downloaded at: https://www.mdpi.com/article/10.3390/agriculture13020301/s1, Figure S1: Morphological characteristics of the other 20 isolates of fungi; Figure S2: Symptoms of leaf spot disease of *Rehmannia glutinosa* in the field.

**Author Contributions:** Conceptualization, X.L. (Xiangyang Liu) and X.Y.; methodology, T.D., X.S., Y.C., Y.Z. and X.L. (Xiangyang Liu); investigation, F.G., X.S. and Z.Z.; data curation, T.D., J.D., F.W., X.L. (Xuanzhen Li) and X.L. (Xiangyang Liu); writing—original draft preparation, T.D., X.S., Y.C., F.G. and J.D.; writing—review and editing, T.D., Y.C., S.A. and X.L. (Xiangyang Liu). All authors have read and agreed to the published version of the manuscript.

**Funding:** This research was funded by the Key Scientific Research Projects of Henan Province (21B210002) and the Science and Technology Innovation Fund Project of Henan Agricultural University (KJCX2020A13), the Earmarked Fund for China Agriculture Research System (CARS-27), the National Key Research and Development Program of China (2017YF1700705).

**Institutional Review Board Statement:** Not applicable.

**Data Availability Statement:** Not applicable.

**Acknowledgments:** We thank Ying Zhao, Yuehua Geng, Shengli Ding, and Bingjian Sun at College of Plant Protection, Henan Agricultural University for their help in pathogen identification and preservation.

**Conflicts of Interest:** The authors declare no conflict of interest.

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
