# Peer review of "Characterization and Fungicide Screening of a New Pathogen That Causes Leaf Spot on Rehmannia glutinosa"

_agriculture, doi:10.3390/agriculture13020301_

Round 1

Reviewer 1 Report

The English is very poor and needs a re-check thoroughly by a native English speaker or by a professional Language Editing Service. Besides the grammatical errors, the sentences need to recheck scientifically as there is no clarity.

1. The abstract of manuscript need to rewrite considering the clarity of scope, importance of the disease, background, aim, methodology and results.

2. The technical issues such as line 18, P-1, “ how can be 25 isolates were isolated” , should be addressed and rephrased properly.

3. The key words should be re write following the research paper guideline i.e. key words should be different from the words used in title.

4. Insert a space between the value and its unit throughout the MS.
Example: 11 °C is correct. (But no space between the value and %. Example: 11% is correct.)

5. What was the standard methods followed to prepare the media neither described nor any reference cited ? Instead of the text, the name and the composition of the different media used in this study may be presented in tabular form and similarly the fungicides too.

6. Rephrase the line 97-98 on P-3  Infected leaf samples were washed and cleaned with water and 97 then dried with clean blotting paper.

7. The authors mentioned that they used 4 day old culture for establishing the pathogenicity of isolates, fusaria is slow growing and mostly took 7 day for proper growth and conidia, why they used only 4 days old culture.??

8. On P-4 line 151, heading of the para need to redefine as no biological system used to characterize the fungal isolates.

Also see the comments in attached file.

Reviewer 2 Report

In the publication "Characterization and fungicide screening of a new pathogen that causes leaf spot on Rehmannia glutinosa," the authors describe and screen a new pathogen that causes Rehmannia glutinosa leaf spots.

#From the introduction to discussion, the study is well done. The pathogen affects cereals and can cause illnesses such root rot, stem rot, and head blight. Additionally, it affects harvests of vegetables and fruits, causing disease including mango leaf spots, coriander stem and root rot, postharvest Fusarium rot of mandarins, including Aleppo pine and Avicennia marina.

#Chinese medicinal plants are also reportedly affected by F. equiseti, according to the authors. This study documents the presence of leaf spot on R. glutinosa brought on by F. equiseti for the first time. They documented the signs and symptoms of leaf spot and used morphological and genetic traits to pinpoint the pathogen. The biological traits of F. equiseti, which was isolated from R. glutinosa, were then investigated.

#Eight fungicides' antifungal activity against R. glutinosa were also identified. The bioassays revealed that the eight fungicides that were chosen had specific antifungal effects on F. equiseti and F. acuminatum, and the same fungicide had very similar antifungal levels on the two species of the Fusarium genus. Therefore, it is possible to speculate that the data may be useful for controlling the majority of Fusarium genus species.

#I have a few reservations, despite the fact that the study is well-structured and thoroughly explained:

I have a few reservations, despite the fact that the study is well-structured and thoroughly explained: In order to convey a message to the reader throughout the entire document, sentences may be rephrased.

#  The discussion should be revised, with the focus of the conversation shifting to the hypothesis, future goals, and all alone the appropriateness of the discussion.

#English should be improved throughout the study.

Because of the above issues, I advise extensive revision prior to acceptance.

Reviewer 3 Report

Dear Authors,

Despite the time and effort invested in to paper, there are severe issues that must be addressed. English should be checked, and writing must be more concise. Because the sequences of the chosen isolates are not available in the NCBI gene bank, the molecular aspect cannot be evaluated. It is unclear why the author's previous study isolates were included in the analyses. Titles and content must be more specific and less ambiguous. The introduction and discussion should be improved so that they explain the topic, the significance, and the position of this results within the available literature.

Round 2

Reviewer 1 Report

The authors attempted enough morphological as well as molecular keys for charcterization of the associated fungal pathogen.  

Reviewer 2 Report

Authors have revised manuscript intensively and now manuscript is acceptable in its present form.